# Ca^2+^ Microdomains, Calcineurin and the Regulation of Gene Transcription

**DOI:** 10.3390/cells10040875

**Published:** 2021-04-12

**Authors:** Gerald Thiel, Tobias Schmidt, Oliver G. Rössler

**Affiliations:** Department of Medical Biochemistry and Molecular Biology, Saarland University Medical Faculty, D-66421 Homburg, Germany; Schmidttobias190@gmail.com (T.S.); oliver.roessler@uks.eu (O.G.R.)

**Keywords:** Ca^2+^ microdomains, Ca^2+^ channel, calcineurin, CREB, Elk-1, NFAT, TRPM3

## Abstract

Ca^2+^ ions function as second messengers regulating many intracellular events, including neurotransmitter release, exocytosis, muscle contraction, metabolism and gene transcription. Cells of a multicellular organism express a variety of cell-surface receptors and channels that trigger an increase of the intracellular Ca^2+^ concentration upon stimulation. The elevated Ca^2+^ concentration is not uniformly distributed within the cytoplasm but is organized in subcellular microdomains with high and low concentrations of Ca^2+^ at different locations in the cell. Ca^2+^ ions are stored and released by intracellular organelles that change the concentration and distribution of Ca^2+^ ions. A major function of the rise in intracellular Ca^2+^ is the change of the genetic expression pattern of the cell via the activation of Ca^2+^-responsive transcription factors. It has been proposed that Ca^2+^-responsive transcription factors are differently affected by a rise in cytoplasmic versus nuclear Ca^2+^. Moreover, it has been suggested that the mode of entry determines whether an influx of Ca^2+^ leads to the stimulation of gene transcription. A rise in cytoplasmic Ca^2+^ induces an intracellular signaling cascade, involving the activation of the Ca^2+^/calmodulin-dependent protein phosphatase calcineurin and various protein kinases (protein kinase C, extracellular signal-regulated protein kinase, Ca^2+^/calmodulin-dependent protein kinases). In this review article, we discuss the concept of gene regulation via elevated Ca^2+^ concentration in the cytoplasm and the nucleus, the role of Ca^2+^ entry and the role of enzymes as signal transducers. We give particular emphasis to the regulation of gene transcription by calcineurin, linking protein dephosphorylation with Ca^2+^ signaling and gene expression.

## 1. Introduction

Cells express a variety of surface receptors and ion channels that regulate intracellular Ca^2+^ ion concentration. At rest, the intracellular cytosolic Ca^2+^ concentration is estimated to be at 100 nM while the extracellular milieu has a Ca^2+^ concentration of approximately 1 mM. Thus, opening of voltage-gated or ligand-gated Ca^2+^ channels in the plasma membrane leads to an influx of Ca^2+^ ions into the cell. In addition, stimulation of Gαq-coupled receptors triggers the activation of intracellularly located IP_3_ receptors and results in an influx of Ca^2+^ ions from the endoplasmic reticulum into the cytosol (Figure 1). This drop in the Ca^2+^ concentration of the endoplasmic reticulum is sensed by the STIM proteins of the endoplasmic reticulum. They oligomerize and contact Orai Ca^2+^ channels of the plasma membrane, leading to an influx of Ca^2+^ ions from the outside into the cell. This scenario is known as store-operated Ca^2+^ entry (SOCE). Even lysosomes, often viewed as a “terminal end” of intracellular pathways, are involved in intracellular Ca^2+^ signaling [1]. Ca^2+^ ions are second messengers and activate various Ca^2+^-responsive protein kinases and calcineurin, a Ca^2+^/calmodulin-dependent protein phosphatase, in the cytosol. These enzymes function as signal transducers for propagating the Ca^2+^ signal from the cytosol into the nucleus. Amplitude, duration and subcellular localization of Ca^2+^ signals can differ depending on the nature of the receptor or channel and these Ca^2+^ signals can be modified by Ca^2+^-accumulating and Ca^2+^-releasing organelles, Ca^2+^ pumps and Ca^2+^ exchangers.

## 2. Signal Transducers Required for Ca^2+^-Induced Signaling from the Plasma Membrane to the Nucleus

A rise in intracellular Ca^2+^ leads to the activation of signal transducers that propagate the Ca^2+^ signal into the nucleus, as depicted in Figure 1. The activation of the mitogen-activated protein (MAP) kinase extracellular signal-regulated protein kinase (ERK1/2) is well studied. MAP kinases, a group of structurally related serine/threonine protein kinases, are activated by many extracellular signaling molecules including mitogens, growth factors, cytokines, metabolites, ligands of G protein-coupled receptors and Ca^2+^ ions. MAP kinases are part of a phosphorylation signaling cascade involving two upstream kinases called MAP kinase kinase (or MAP2 kinase) and MAP kinase kinase kinase (or MAP3 kinase) [2]. ERK1/2 is activated by a rise in intracellular Ca^2+^ [3,4,5,6,7,8,9,10,11], most likely involving the protein kinases PKC and Raf. The key role of ERK1/2 as a cytoplasmic signal transducer connecting a rise in cytoplasmic Ca^2+^ and the activation of gene transcription has been confirmed by pharmacological or genetic inhibition of the ERK1/2 signaling cascade [10,12,13,14,15], or overexpression experiments involving MAP kinase phosphatase-1 and DA-Raf-1 [8,9,16,17], a splice form of A-Raf that functions as a dominant negative inhibitor of Raf. The MAP kinases c-Jun *N*-terminal protein kinase (JNK) and p38 have also been identified as signal transducers [10,18,19]. These kinases are activated by growth factors and proinflammatory cytokines, but also by stress signals (oxidative stress, UV light). MAP kinases translocate into the nucleus upon phosphorylation and catalyze the phosphorylation of nuclear substrates, including Ca^2+^-responsive transcription factors. A rise in cytoplasmic Ca^2+^ may also lead to an activation of Ca^2+^/calmodulin-dependent protein kinases that phosphorylate and activate Ca^2+^-responsive transcription factors [20,21]. Moreover, cytoplasmic Ca^2+^ ions are required to activate the Ca^2+^/calmodulin-dependent phosphatase calcineurin that transduces the Ca^2+^ signal into the nucleus via dephosphorylation of cytoplasmic transcription factors.

It has been suggested that the route of Ca^2+^ entry, either via ligand- or voltage-gated Ca^2+^ channels, plays a key role in determining whether a particular transcription factor is activated or not [22]. We think that it is instead the amplitude and duration of Ca^2+^ signal, controlled by the expression levels of particular Ca^2+^ channels and the nature and concentration of the ligands, that are of major importance to Ca^2+^-induced gene transcription. Stimulation of the cation channel TRPM3 by either pregnenolone sulfate or nifedipine leads to an increase in cytoplasmic Ca^2+^, as measured by fluorometric Ca^2+^ imaging [23]. However, only stimulation with pregnenolone sulfate triggered Ca^2+^-dependent activation of gene transcription [24], indicating that Ca^2+^ influx into the cells alone does not necessarily lead to a Ca^2+^-dependent change in gene transcription. Regulation of gene transcription has been shown to require sustained Ca^2+^ signals, whereas transient Ca^2+^ signals may be insufficient to trigger transcriptional responses [25]. In addition, Ca^2+^-responsive transcription factors have different sensitivities to Ca^2+^ signals. Low levels of Orai channel stimulation led to an activation of nuclear factor of activated T-cells (NFAT), while c-Fos activation required increased stimulus intensity [26]. Likewise, the amplitude and the duration of Ca^2+^ signals controls whether NF-κB or NFAT transcription factors are activated in B lymphocytes. While NFAT1 is activated by a low, sustained Ca^2+^ plateau, activation of NF-κB requires a large transient Ca^2+^ rise [27]. Particular Ca^2+^ microdomains may be important to decide whether a Ca^2+^-responsive transcription factor is activated or not. Nuclear translocation of NFAT1, for example, is tightly linked to the Ca^2+^ microdomain generated by the Ca^2+^ influx via Orai channels [28]. Sensitivity to Ca^2+^ signals can vary even within a family of transcription factors. Nuclear translocation of NFAT1 was shown to be more than 5-fold more sensitive to the stimulation of a Gαq-coupled receptor than NFAT4 [28]. In summary, regulation of gene transcription by Ca^2+^ is a “multi-modal system” [29], integrating a variety of signal transducers and transcription factors.

## 3. Ca^2+^-Responsive Transcription Factors

The stimulation of transcription as a result of elevated intracellular Ca^2+^ levels relies on the activation of Ca^2+^-responsive transcription factors. CREB (cyclic AMP-response element (CRE)-binding protein) is the best-known transcription factor that is activated by elevated Ca^2+^ levels. CREB is a basic region-leucine zipper transcription factor with the leucine zipper domain functioning as the dimerization domain and the basic region as the DNA binding domain (Figure 2A). Originally discovered as a cAMP-responsive transcription factor that binds to the cyclic AMP-response element (CRE), CREB has been shown to be activated by a rise in intracellular Ca^2+^ [20,27,30,31]. Thus, the DNA binding site for CREB, the cyclic AMP-response element, has also been called the Ca^2+^-responsive element [30]. CREB is a substrate for multiple protein kinases including the cAMP-dependent protein kinase PKA, the Ca^2+^/calmodulin regulated protein kinase CaMKIV, mitogen and stress-induced protein kinase (MSK), MAP kinase kinase 6, protein kinase B, ribosomal S6 kinase and others [32,33,34,35].

A rise in intracellular Ca^2+^ activates the transcription factor Elk-1 that belongs to the family of ternary complex factors. Together with a dimer of the serum response factor (SRF), Elk-1 binds to the serum-response element (SRE) (Figure 2B) [36,37]. Elk-1 has an *N*-terminal DNA binding domain, a *C*-terminal transcriptional activation domain and an SRF-interaction domain (B domain), necessary for the formation of the ternary Elk-1-SRF complex (Figure 2C). Elk-1 is phosphorylated by the MAP kinases ERK1/2, p38 and JNK [38,39,40,41], connecting intracellular signaling cascades with the activation of SRE-mediated transcription.

NFAT activation is also triggered by a rise in intracellular Ca^2+^. This family of proteins is involved in the regulation of developmental programs of different vertebrate tissues [42]. The NFAT proteins have an *N*-terminal transcriptional activation domain (TAD), a regulatory domain (NFAT homology region, NHR) and a *C*-terminal DNA-binding domain (Rel-homology domain, RHR. The NHR contains binding sites for protein kinases (casein kinase-1, glycogen synthase kinase-3, dual-specific tyrosine-phosphorylation-regulated kinase DYRK) and for the protein phosphatase calcineurin (Figure 2D).

## 4. Ca^2+^-Induced Gene Transcription: Role of Nuclear Ca^2+^

There has been debate about the role of nuclear Ca^2+^ in Ca^2+^-induced gene transcription. It has been suggested that the activation of Ca^2+^-responsive transcription factors does not solely rely on the activation of Ca^2+^-responsive signal transducers such as protein kinases and phosphatases. Rather, it has been proposed that the elevation of nuclear Ca^2+^ directly contributes to the activation of the Ca^2+^-responsive transcription factors. This hypothesis is based on the assumption that a rise of cytoplasmic Ca^2+^ is directly propagated to the nucleus via diffusion of Ca^2+^ ions through the nuclear pore into the nucleus. Ca^2+^ signals, generated by neuronal activity at the plasma membrane, have been shown to unrestrictedly cross the nuclear pore in neurons [43]. However, changes in the nuclear Ca^2+^ concentration reflect changes in cytoplasmic Ca^2+^. Therefore, it is difficult to distinguish between nuclear or cytoplasmic Ca^2+^ pools. In addition, the amplitude and the duration of Ca^2+^ signals are further modified by Ca^2+^-accumulating organelles such as mitochondria and the endoplasmic reticulum expressing Ca^2+^ pumps and Ca^2+^ exchange proteins. This suggests that the Ca^2+^ signal that reaches the nucleus is reduced in comparison to the signal originally generated in the cytoplasm. An alternative hypothesis is that nuclear Ca^2+^ is increased independent of cytoplasmic Ca^2+^ via the activation of nuclear IP_3_ receptors that release Ca^2+^ ions directly into the nucleoplasm upon stimulation with IP_3_. Thus, IP_3_, that is generated in the cytoplasm, following activation of Gαq-coupled receptors or receptor tyrosine kinases, may diffuse into the nucleus and may activate nuclear IP_3_ receptors that are expressed in the inner leaflet of the nuclear membrane, facing the nucleoplasm [44,45,46]. However, the presence of IP_3_ receptors at this location is controversial [46].

Despite the ongoing debate over the role of nuclear Ca^2+^, several transcription factors that are responsive to changes in Ca^2+^ concentration have been identified. CREB has been called a “nuclear calcium-responsive transcription factor” [47], based on experiments involving activation of L-type voltage-gated Ca^2+^ channels in AtT20 pituitary cells and microinjection of a Ca^2+^ chelator into the nucleus to block CREB-regulated gene transcription. Surprisingly, the role of cytoplasmic signal transducers was not analyzed in this study. In contrast, an investigation of glucose stimulation of INS-1 insulinoma cells revealed that glucose led to an activation of CREB-regulated gene transcription, involving an influx of Ca^2+^ ions into the cytoplasm of the cells and an increase in cytoplasmic Ca^2+^ (Figure 3A). Pharmacological experiments showed that the glucose/Ca^2+^-induced signaling cascade required the activation of l-type voltage-gated Ca^2+^ channels (Figure 3B) and ERK1/2 as a signal transducer (Figure 3C) [48]. The ERK1/2 signaling pathway was blocked with the compound PD98059 that interfered with the phosphorylation of the protein kinase MEK by Raf. Activation of ERK1/2 as a result of glucose stimulation and l-type voltage-dependent Ca^2+^ channel activation of insulinoma cells has also been observed by other investigators [5]. In the experiments addressing the role of nuclear Ca^2+^ for CREB-regulated gene transcription in AtT20 cells, the strategy of injecting a Ca^2+^ buffer into the nucleus of single cells was used [47]. This strategy has the disadvantage that it is not compatible with performing biochemical analyses. As an alternate, targeted expression of the Ca^2+^ binding protein, parvalbumin in either the nucleus or the cytosol has been successfully used to buffer Ca^2+^ signals in distinct compartments [14,48,49]. Parvalbumin is a low-molecular-weight protein that binds Ca^2+^ with high affinity via two EF-hand motifs [50]. High resolution recording of subcellular Ca^2+^ transients revealed that cytosolic and nucleoplasmic expression of the Ca^2+^ buffer parvalbumin specifically reduced the amplitude of Ca^2+^ transients in the respective subcellular compartment [48]. Moreover, biochemical analysis revealed that glucose-induced transcription of a CREB-regulated reporter gene, through binding to CREs in the promoter region, required a rise in cytoplasmic Ca^2+^, while buffering of the nuclear Ca^2+^ did not change the transcription of the reporter gene (Figure 3D) [48]. Thus, Ca^2+^-induced stimulation of the ERK1/2 signaling pathway is sufficient to stimulate CRE-mediated transcription. A rise in cytoplasmic Ca^2+^ and not a rise in nuclear Ca^2+^ was also shown to be essential for the glucose-triggered activation of transcription factors AP-1 and Elk-1 [48]. In all cases, the activation of the Raf-MEK-ERK1/2 signaling cascade was essential for connecting a rise in cytoplasmic Ca^2+^ with gene transcription. Likewise, stimulation of l-type Ca^2+^ channels in neurons promoted an activation of ERK1/2 that in turn led to the phosphorylation of CREB and the transcription of CREB target genes [4].

In addition to CREB, the ternary complex factor Elk-1 has been described to be regulated by the nuclear Ca^2+^ concentration following stimulation of HepG2 hepatoma cells or HEK293 cells with EGF [49]. Stimulation of the EGF receptor may lead to an increase in cytoplasmic Ca^2+^ via stimulation of phospholipase Cγ, but also triggers a strong activation of the ERK1/2 signaling cascade [51]. It has been shown that stimulation of the cation channel TRPM3 leads to the increased transcriptional activation potential of Elk-1 involving an influx of Ca^2+^ ions into the cells and the activation of ERK1/2 (Figure 4A, B) [14]. Experiments using genetically encoded Ca^2+^ buffers revealed that a rise in cytoplasmic Ca^2+^ is required for the upregulation of the transcriptional activation potential of Elk-1 (Figure 4C) [14]. In contrast, buffering nuclear Ca^2+^ had no inhibitory effect on the transcriptional activity of Elk-1. Thus, Ca^2+^-induced activation of Elk-1 required a rise in cytoplasmic Ca^2+^ and the subsequent stimulation of the Raf-MEK-ERK1/2 signaling pathway and was independent of the nuclear Ca^2+^ concentration. The observation that a rise in cytoplasmic Ca^2+^ and not in nuclear Ca^2+^ is essential for the activation of Elk-1 via TRPM3 is corroborated by experiments involving inhibition of the cytoplasmic enzyme MAP kinase kinase with the compound PD98059 [14]. Inhibition of phosphorylation of MAP kinase kinase blocks the phosphorylation and activation of ERK1/2, the key enzyme of the ERK1/2 signaling pathway.

NFAT transcription factors are cytoplasmic phosphoproteins that migrate into the nucleus following dephosphorylation catalyzed by calcineurin. While NFAT1 remains in the nucleus for some time, NFAT4 is rapidly exported out of the nucleus. A sophisticated study employing the nuclear-directed Ca^2+^ buffer parvalbumin revealed that NFAT4 retention in the nucleus was dependent on a high level of Ca^2+^. In contrast, the presence of NFAT1 in the nucleus was independent of the Ca^2+^ concentration in the nucleus [28]. Viewing the entire signal cascade, these results suggest that a high stimulus intensity triggers the nuclear translocation of both NFAT1 and NFAT4 and the sustained expression of both proteins in the nucleus, whereas a low stimulus intensity also triggers NFAT1 and NFAT4 translocation, but NFAT4 is rapidly exported back into the cytoplasm. By this means, the stimulus intensity may regulate different genetic patterns, dependent either on both NFAT1 and NFAT4 (high intensity) or only on NFAT1 (low intensity). Moreover, using a genetically encoded nuclear IP_3_ buffer, it was shown that nuclear expression of NFAT4 requires the activation of nuclear IP_3_ receptors by IP_3_ that had been generated at the plasma membrane and diffused into the nucleus. As a result, the Ca^2+^ concentration in the nucleus increased even when cytoplasmic Ca^2+^ ions were buffered [45], prolonging the nuclear presence of NFAT4. However, we have to point out that the investigators measured translocation of NFAT proteins in this study, but did not directly measure NFAT-regulated gene transcription. NFAT proteins bind DNA with relatively low affinity. Formation of a complex with a nuclear partner strongly increases their DNA affinity. In particular, NFAT proteins associate with the transcription factor AP-1. Thus, sustained activity of NFAT4 in the nucleus would require a sustained nuclear expression of an NFAT4 binding partner. Thus, without identification of the nuclear partner, the sustained presence of NFAT4 in the nucleus alone does not necessarily predict sustained NFAT4-controlled gene transcription. The hypothesis of Kar et al. [45], that the concentration of nuclear Ca^2+^ induces distinct genetic signatures by regulating NFAT nuclear retention or export, will require proof that the translocated NFAT proteins are able to activate common and distinct genes. Finally, the presence of nuclear calcineurin that would be necessary to keep nuclear NFAT proteins in a dephosphorylated state has not been investigated in this study.

Whether there is a distinct role for nuclear Ca^2+^ in gene transcription is still contentious. The use of a genetically encoded Ca^2+^ buffer has certainly helped in elucidating the role of cytoplasmic versus nuclear Ca^2+^. Importantly, the role of ERK1/2 as signal transducer that translates high cytoplasmic Ca^2+^ levels into the activation of Ca^2+^-regulated transcription factors in the nucleus has to be considered. The investigation of NFAT1 and NFAT4 translocation provided further evidence for the role of cytoplasmic, and not solely nuclear, Ca^2+^ in activation of gene transcription [52].

## 5. Calcineurin

The Ca^2+^/calmodulin-dependent protein phosphatase calcineurin is a heterodimer, composed of the catalytic calcineurin A-subunit and the regulatory B-subunit. The modular structure of calcineurin A, depicted in Figure 5A, shows that the protein has an extended catalytic domain, followed by a binding site for calcineurin B, a calmodulin-binding domain and a C-terminal autoinhibitory domain. The active site of the enzyme has a dinuclear (Fe^3+^-Zn^2+^) metal center. In the absence of Ca^2+^ ions, calcineurin is inactive, due to a binding of the autoinhibitory domain to the catalytic center. An influx of Ca^2+^ ions from the external milieu into the cytosol (via activation of cell-surface-located Ca^2+^ channels) or an influx of Ca^2+^ ions from the endoplasmic reticulum into the cytosol (following stimulation of Gαq-coupled receptors) leads to the activation of calcineurin. The regulatory B-subunit is tightly bound to calcineurin A, whereas calmodulin binding is Ca^2+^ dependent. Activation of calcineurin requires the binding of Ca^2+^ ions to calcineurin B and the Ca^2+^-dependent binding of a Ca^2+^/calmodulin complex to calcineurin A. The binding of calmodulin to the calmodulin-binding domain triggers the displacement of the autoinhibitory domain, leading to the activation of the enzyme [53,54,55,56]. A truncated calcineurin A mutant, called ΔCnA (Figure 5B), lacks the *C*-terminal autoinhibitory domain and a portion of the calmodulin binding site. It is constitutively active and does not need Ca^2+^ ions for its activation. The small regulatory calcineurin B-subunit (Figure 5C) shows homology with the Ca^2+^ binding protein calmodulin (Figure 5D) and binds Ca^2+^ ions via four EF-hand Ca^2+^ binding motifs.

Calcineurin is ubiquitously expressed, with the highest expression in the brain. The enzyme is mainly found in the cytoplasm, but calcineurin immunoreactivity has also been found in the nucleus [28,55,57]. Calcineurin has been shown to migrate into the nucleus in thapsigargin-stimulated RBL mast cells [28]. Expression of a fusion protein of the truncated calcineurin A mutant ΔCnA with EGFP showed that the enzyme was in the cytoplasm (10), suggesting that cytoplasmic calcineurin affects gene transcription. Thus, calcineurin is activated following a rise in cytoplasmic Ca^2+^, either by activating Ca^2+^ channels of the plasma membrane or the endoplasmic reticulum, or by the release of Ca^2+^ ions from intracellular organelles such as mitochondria or lysosomes.

Calcineurin became of medical interest following the discovery that the clinically important immunosuppressive drugs tacrolimus (FK506) and cyclosporin A, used for the treatment of organ rejection in transplanted patients, function as potent calcineurin inhibitors when they are bound to the immunophilins FKBP12 or cyclophilin A, respectively [58]. However, we would like to emphasize that neither tacrolimus nor cyclosporin A is calcineurin-specific. The immunophilins are cytosolic peptidyl-prolyl isomerases that are involved in the regulation of many biological functions, including ion channel activation, intracellular Ca^2+^ release and protein folding [59,60]. In addition to regulating calcineurin activity, cyclosporin A has been described as an inhibitor of mitochondrial Ca^2+^ uptake and the mitochondrial permeability transition pore [61,62]. Cyclosporin A also modulates innate immunity [63]. Tacrolimus has been identified as an activator of the nonselective cation channel TRPM8 [64] that is primarily stimulated by cold temperature and cooling-agents such as menthol or icilin. Many investigators have used either tacrolimus or cyclosporin A as a calcineurin inhibitor. However, puzzling data have sometimes been published which may be explained by the other activities of these pharmacological compounds. There are several genetically encoded calcineurin inhibitors that may be more suitable for investigation of calcineurin functions. The regulator of calcineurin (RCAN) proteins inhibit calcineurin activity very efficiently and have been shown to inhibit calcineurin-dependent gene transcription [65]. RCAN1 binds to a site within calcineurin A, located between the catalytic domain and the binding site for the regulatory calcineurin B subunit. The RCAN1-encoding gene is located on chromosome 21. Overexpression of RCAN1 has been found in the brain of Alzheimer’s disease patients and humans with Down syndrome. The RCAN proteins are thought to establish a negative feedback loop that prevents high calcineurin activity [65,66].

## 6. Calcineurin Regulates Gene Transcription

Calcineurin-induced dephosphorylation of proteins is connected with several biological functions, including inhibition of neurotransmission, desensitization of postsynaptic Ca^2+^ channels, activation of nitric oxide synthase, and regulation of the Na^+^, K^+^-ATPase and others. Here, we focus our attention on the regulation of gene transcription by calcineurin-mediated protein dephosphorylation.

### 6.1. Calcineurin-Catalyzed Dephosphorylation Activates NFAT

The best studied substrates of calcineurin are the transcription factors of the NFAT family that are regulated by the cytoplasmic Ca^2+^ concentration. Calcineurin is activated by a rise in cytosolic Ca^2+^ concentration, which can be generated in various microdomains of the cell. A rise in cytoplasmic Ca^2+^ may be the result of a Ca^2+^ influx through Ca^2+^ channels of the plasma membrane (voltage-gated Ca^2+^ channels, Orai channels) or the endoplasmic reticulum (ionotropic IP_3_ receptors) [28,67,68]. Calcineurin may also be activated by a release of Ca^2+^ ions from mitochondria. Prolonged Ca^2+^ release from the mitochondria of sensory neurons facilitates nuclear transport of NFAT in neurons [69]. The NFAT proteins 1–4 are cytoplasmic phosphoproteins that translocate from the cytoplasm to the nucleus upon dephosphorylation by calcineurin [42,54,56,70,71]. NFAT proteins return to the cytoplasm following rephosphorylation by constitutively active protein kinases, indicating that the balance between NFAT kinases and calcineurin determines whether NFAT proteins translocate from the cytoplasm into the nucleus. Moreover, sustained NFAT transcriptional activity requires prolonged expression of calcineurin in the nucleus and sustained elevated nuclear Ca^2+^ levels. Figure 6A shows that expression of a constitutively active form of calcineurin A is sufficient to activate transcription of a chromatin-embedded NFAT-responsive reporter gene [68]. There are two binding sites for calcineurin in the regulatory domains of NFAT proteins. The major calcineurin binding site A, located at the *N*-terminus of the regulatory region, encompasses the consensus sequence PxIxIT, as depicted in Figure 2D. Next to the calcineurin binding site A is the “serine-rich gatekeeper region” (SRR-1) with several phosphorylated serine residues. The nuclear localization signal (NLS) is also located within the regulatory domain of NFAT proteins and is masked by the phosphate groups, so that NFAT remains in the cytoplasm. Calcineurin-catalyzed dephosphorylation of NFAT proteins exposes the NLS to the nuclear translocation machinery, so that NFAT migrates into the nucleus. Mutation of the calcineurin binding site A impairs dephosphorylation and nuclear translocation of NFAT proteins [72,73]. Expression of a peptide spanning the calcineurin binding site attenuated NFAT dephosphorylation, NFAT nuclear translocation, and NFAT-controlled gene transcription [72]. Based on the calcineurin docking site to NFAT, a peptide sequence termed VIVIT was identified from combinatorial peptide libraries showing a much higher affinity to calcineurin than the original calcineurin binding site. The peptide inhibited NFAT activation and NFAT-regulated transcription of cytokine-encoding genes in T cells. In the nervous system, expression of the VIVIT peptide inhibited NFAT activation and alleviated amyloid beta neurotoxicity [74,75]. Exchange of the PRIEIT sequence in NFAT1 with the PVIVIT generated an NFAT1 molecule that was dephosphorylated under resting conditions and was partially found in the nucleus [74]. These data suggest that the increased affinity of calcineurin to the mutated NFAT1 protein allowed a competition between the calcineurin substrate NFAT1 and the autoinhibitory domain of calcineurin for binding to the catalytic site of calcineurin, even under resting Ca^2+^ concentrations.

Activation of NFAT proteins requires a Ca^2+^ rise in the cytoplasm and presence of active calcineurin. The hypothesis that a rise in nuclear Ca^2+^ activates NFAT4 [45] implies that the rise in nuclear Ca^2+^ is accompanied by a rise of activated calcineurin in the nucleus, because NFAT proteins must be kept in a dephosphorylated state to sustain nuclear expression. It has been proposed that calcineurin is transported in the nucleus together with NFAT4 where it continues to dephosphorylate NFAT4 [76]. However, a comparison of the time course of calcineurin and NFAT1 and NFAT4 migration argues against a co-transport of NFAT proteins with calcineurin [28]. It would be interesting to know whether expression of an endogenous calcineurin inhibitor such as RCAN1 in the nucleus would induce export of NFAT4 from the nucleus into the cytoplasm. Moreover, the hypothesis of Kar et al. [45] is built on the assumption that nuclear expression of NFAT proteins can be used as measure for NFAT-regulated gene transcription. To support the hypothesis that nuclear Ca^2+^ activates NFAT4, a direct gene transcription assay is needed to clearly show the upregulation of NFAT-regulated gene transcription under these conditions.

### 6.2. Calcineurin Modulates CREB Activity

NFAT transcription factors are active in the dephosphorylated state. Thus, dephosphorylation by calcineurin induces activation of NFAT-controlled transcription. In contrast, several transcription factors are activated by phosphorylation. Accordingly, dephosphorylation is connected with inactivation of gene transcription by these transcription factors. The prototype of a phosphorylation-regulated transcription factor is CREB. Phosphorylation by several protein kinases, in particular PKA and Ca^2+^/calmodulin-dependent protein kinase IV, activates CREB-regulated transcription, whereas dephosphorylation by protein phosphatases 1 and 2A inactivates the protein. CREB is not a substrate for calcineurin, but is indirectly regulated by this phosphatase. Figure 6B shows that CREB-regulated transcription, induced by stimulating Gαq-coupled designer receptors, was attenuated in the presence of the constitutively active calcineurin mutant ΔCnA. In line with this observation, the transcription activation potential of CREB was reduced in the presence of ΔCnA (Figure 6C). Calcineurin, however, did not reduce the transcriptional activity of the C2/CREB mutant that activated CRE-mediated transcription independently of phosphorylation [68]. These data were corroborated by data showing that CREB-regulated gene transcription is activated following inhibition of calcineurin by RCAN1 expression [77]. These experiments fit very well with the hypothesis that calcineurin functions as a negative regulator of CREB. Calcineurin dephosphorylates inhibitor-1, a protein inhibitor of protein phosphatase-1. Phosphorylation of inhibitor-1 by PKA turns this protein in an active protein phosphatase-1 inhibitor, while the dephosphorylated inhibitor-1 is inactive. Dephosphorylation of inhibitor-1 by calcineurin relieves the inhibition of protein phosphatase-1, resulting in a dephosphorylation and inactivation of CREB [78]. It is interesting that a rise in intracellular Ca^2+^ activates CREB-controlled gene transcription (via activation of Ca^2+^/calmodulin-dependent protein kinase IV) and subsequently triggers an inactivation of CREB (via calcineurin-catalyzed dephosphorylation of inhibitor-1 and the activation of protein phosphatase-1). Calcineurin is therefore part of a negative feedback loop of Ca^2+^-initiated transcription of CREB target genes.

Alternatively, calcineurin has been described as a positive regulator of CREB. According to this scenario, calcineurin dephosphorylates the transducer of regulated CREB activity-2 (TORC2) which is sequestered in the cytoplasm via binding to 14-3-3 proteins. Ca^2+^-activated calcineurin catalyzes the dephosphorylation of TORC2, allowing its translocation into the nucleus. TORC2 binds to the bZIP domain of CREB and functions as a coactivator [79]. Our experiments prefer the first hypothesis with calcineurin as a negative regulator of CREB-mediated transcription. However, both hypotheses have not been analyzed and compared in detail, which would involve an analysis of the phosphorylation state of inhibitor-1 and TORC2, the activity of protein phosphatase-1 and the nuclear expression of these proteins after a rise in cytoplasmic Ca^2+^.

### 6.3. Calcineurin Dephosphorylates Elk-1 and Inhibits SRE-Mediated Transcription

The ternary complex factor protein Elk-1 has been identified as a major substrate for calcineurin [80,81]. Elk-1 requires phosphorylation of sites within the transcriptional activation domain, primarily on serine residue 383, to become an active transcription factor. Accordingly, dephosphorylation of Elk-1 is connected with inhibition of Elk-1-mediated gene transcription. In vitro experiments showed that calcineurin dephosphorylates Elk-1 in the presence of Ca^2+^ and calmodulin [80]. Using phospho-specific antibodies, dephosphorylation of serine residue 383 of Elk-1 by calcineurin was demonstrated [81]. Figure 6D shows that the transcriptional activation potential of Elk-1 was significantly reduced in the presence of the constitutively activate calcineurin mutant ΔCnA. Similarly, expression of a constitutively active mutant of calcineurin blocked Raf-1 protein kinase-induced upregulation of the transcriptional activation potential of Elk-1, but did not affect the transcriptional activation potential of the transcription factor ATF2 which is not a substrate for calcineurin [81]. Likewise, stimulus-induced expression of the transcription factor c-Fos was strongly reduced when ΔCnA was overexpressed (Figure 6E). The Egr-1 and c-Fos promoter activities were reduced as well in the presence of ΔCnA [68]. In neurons, it has been shown that expression of the constitutively active mutant of calcineurin A inhibited depolarization-induced expression of c-Fos expression [82].

The connection between calcineurin activity and Elk-1-regulated gene transcription was highlighted in several transgenic mouse models (Table 1). It has been shown that transgenic mice expressing a constitutively active mutant of calcineurin in pancreatic β-cells had decreased β-cell mass, exhibited hyperglycemia and showed enhanced apoptosis [83]. As Elk-1 is a major substrate for calcineurin, expression of the calcineurin mutant in β-cells of transgenic mice induced dephosphorylation and inactivation of Elk-1. This assumption was verified in the analysis of transgenic mice expressing a dominant-negative mutant of Elk-1 (termed REST/Elk-1ΔC) in pancreatic β-cells [84]. The morphometric analysis of these mice revealed that the islets were approximately 50% smaller when the transgene was expressed. These data indicate that an Elk-1-regulated gene expression program is required for the generation of islets of adequate size. Experiments performed with REST/Elk-1ΔC-expressing mice showed that transgene expression resulted in increased caspase-3 activity and impaired glucose tolerance (Table 1 and Figure 7). Thus, expression of a dominant-negative mutant of Elk-1 in pancreatic β-cells resulted in a similar phenotype as the expression of a constitutively active mutant of calcineurin. A major target of Elk-1 is the Egr-1 gene, encoding a zinc finger transcription factor. Expression of the dominant-negative mutant of Elk-1 almost completely blocked the stimulus-induced activation of Egr-1 biosynthesis in insulinoma cells [84]. Interestingly, transgenic mice expressing a dominant-negative mutant of Egr-1 in pancreatic β-cells also had smaller pancreatic islets in comparison to control mice, increased caspase-3/7 activities and were hyperglycemic in glucose tolerance tests (Table 1 and Figure 7) [85]. Further supporting the view that calcineurin functions as a negative regulator of SRE-controlled genes, it has been shown that suppression of calcineurin activity in hippocampal neurons resulted in upregulation of Egr-1 expression [82].

## 7. Conclusions

Ca^2+^ ions function as second messengers in cells and regulate multiple intracellular functions. Stimulus-regulated gene expression is one of the important functions of elevated Ca^2+^ concentrations. Stimulation of plasma membrane receptors or ion channels leads to a rise in intracellular Ca^2+^, but also the release of Ca^2+^ from intracellular stores such as the endoplasmic reticulum, mitochondria and lysosomes. Specialized subcellular Ca^2+^ microdomains orchestrate the specificities of intracellular Ca^2+^ signals. Due to the uptake of Ca^2+^ ions into intracellular Ca^2+^ stores and their export by Ca^2+^ pumps, the duration and periodicity of Ca^2+^ transients may be very different, resulting in different signaling events. Ca^2+^ activates a variety of intracellular enzymes that are used as signal transducers and for prolonging the duration of the Ca^2+^ signal. While the Ca^2+^ concentration may be reduced in the cell by active Ca^2+^ pumps and transporters, the Ca^2+^-activated enzymes remain active and execute the functions initiated by a local Ca^2+^ rise. A particularly interesting Ca^2+^-regulated enzyme is the phosphatase calcineurin that can either activate or inhibit Ca^2+^-responsive transcription factors. Thus, calcineurin links Ca^2+^-induced protein dephosphorylation with the regulation of gene expression.

## Figures and Tables

**Figure 1 cells-10-00875-f001:**
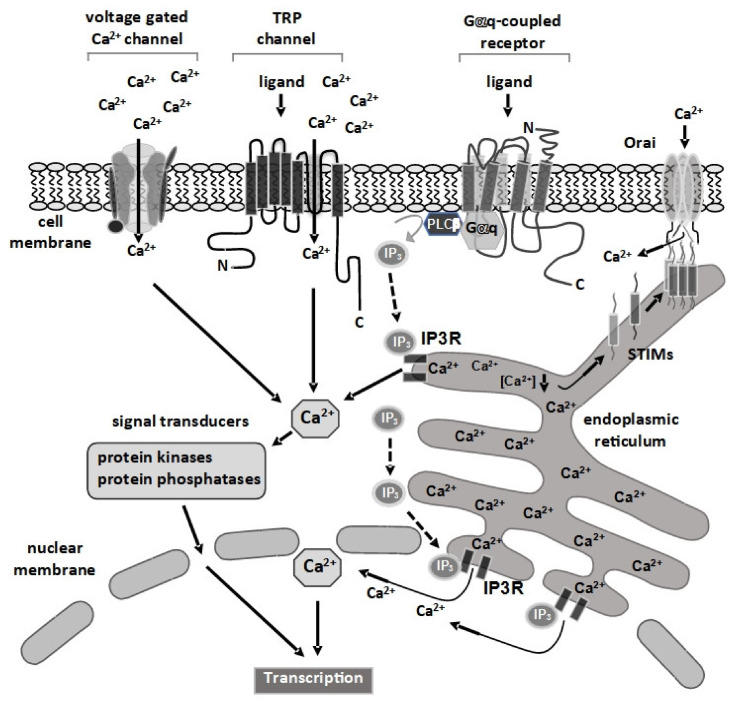
Stimulation of cell surface receptors and channels increases the intracellular Ca^2+^ concentration. Cells express a variety of Ca^2+^ channels at the plasma membrane, including voltage-gated Ca^2+^ channels and ligand-regulated TRP channels. Stimulation of Gαq-coupled receptors activates phospholipase Cβ, leading to the generation of inositol-1,4,5-trisphosphate (IP_3_) and diacylglycerol. IP_3_ is a ligand for the ionotropic IP_3_ receptor, located in the membrane of the endoplasmic reticulum (ER), and binding triggers the influx of Ca^2+^ from the ER store into the cytosol. The depletion of the Ca^2+^ store of the ER is detected by the STIM proteins, Ca^2+^ sensor proteins, that oligomerize and connect with the plasma membrane located Orai Ca^2+^ channels. As a result, Ca^2+^ ions flow into the cell. Since the ER is contiguous with the outer nuclear membrane, it has been proposed that IP_3_ receptors are also expressed in the inner nuclear membrane, facing the nucleoplasm, and their presence allows the release of Ca^2+^ directly into the nucleoplasm. Ca^2+^ ions are second messengers that activate a variety of enzymes in the cytosol, including protein kinases and phosphatases that function as signal transducers. The information is transferred into the nucleus and triggers a change of gene transcription by activating stimulus-responsive transcription factors. Ca^2+^ ions may also diffuse into the nucleus to regulate Ca^2+^-responsive enzymes and transcription factors.

**Figure 2 cells-10-00875-f002:**
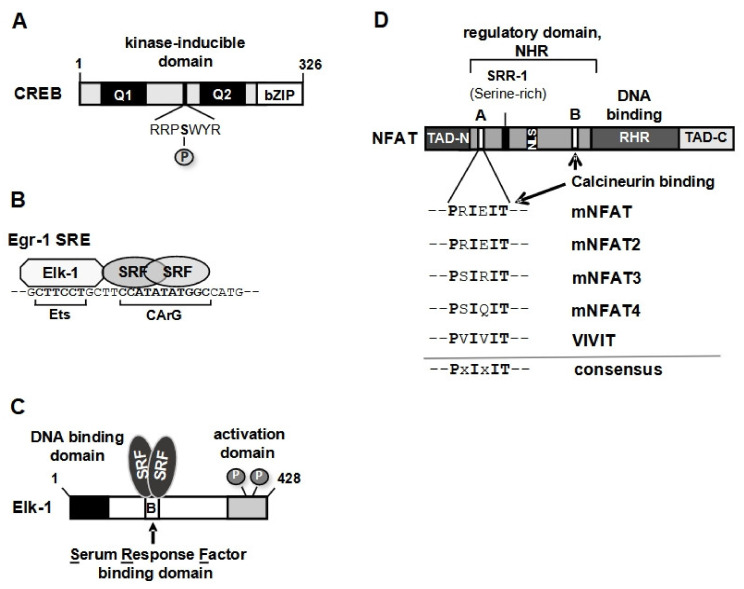
Ca^2+^-responsive transcription factors. (**A**) Modular structure of CREB (cyclic AMP-response element (CRE)-binding protein). The basic region leucine zipper (bZIP) domain, used for dimerization and DNA binding, is located at the *C*-terminus. CREB contains a phosphorylation-dependent transcriptional activation domain (kinase-inducible domain, KID) and two constitutively active glutamine-rich activation domains (Q1 and Q2). The phosphorylation site within the KID is depicted. (**B**) The serum-response element (SRE) provides binding sites for the ternary complex factor Elk-1 and for SRF (serum response factor). SRF binds as a dimer to the consensus sequence CC[A/T]_6_GG, known as CArG box. Elk-1 binds to a DNA motif with the consensus core sequence GGAA/T, called Ets site, because Ets (E26 transformation-specific) transcription factors interact with this DNA sequence. (**C**) Modular structure of Elk-1, showing the *N*-terminal DNA binding domain, the C-terminal activation domain with the key phosphorylation sites and the B domain that is required for interaction with the SRF dimer. (**D**) Modular structure of NFAT transcription factors. The transcriptional activation domains (TAD), the regulatory domain and the DNA binding domain are depicted. The binding sites for calcineurin are indicated. The sequences of the major binding sites of calcineurin to the NFAT proteins, the consensus site and the optimal binding site (VIVIT) are depicted.

**Figure 3 cells-10-00875-f003:**
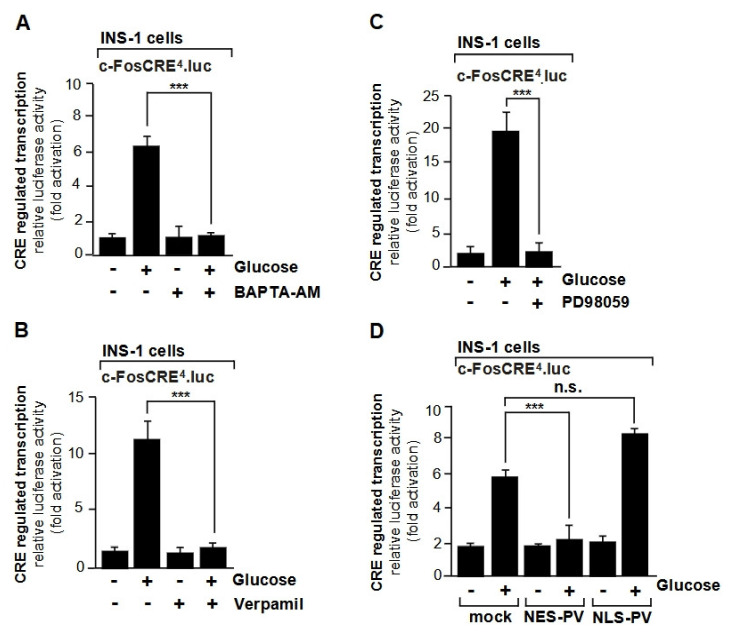
Increased CRE-regulated transcription in glucose-stimulated insulinoma cells requires a rise in cytosolic Ca^2+^. (**A**,**B**,**C**) INS-1 insulinoma cells expressing a chromatin-embedded CRE-responsive reporter gene (c-FosCRE^4^.luc) were maintained in serum-free medium containing 2 mM glucose for 24 h and then stimulated with glucose (11 mM) for 48 h. Cells were cultured in the presence or absence of the Ca^2+^ chelator BAPTA-AM (A), the l-type Ca^2+^ channel inhibitor verapamil (50 µM) (**B**), or the MAP kinase kinase inhibitor (PD98059) (**C**) as indicated. Cell extracts were prepared and analyzed for luciferase activities. Luciferase activity was normalized to the protein concentration. The compound PD98059 blocks phosphorylation of the MAP kinase kinase by Raf, and thus inhibits the ERK1/2 signaling pathway. (**D**) INS-1 insulinoma cells expressing a chromatin-embedded CRE-responsive reporter gene (c-FosCRE^4^.luc) were infected with a lentivirus encoding either mCherry as control, NES-PVmCherry, or NLS-PVmCherry, triggering expression of a parvalbumin-mCherry fusion protein either in the cytosol (NES-PVmCherry) or in the nucleus (NLS-PVmCherry). The cells were maintained in serum-free medium containing 2 mM glucose for 24 h. Stimulation with glucose (11 mM) was performed for 48 h. Cell extracts were prepared and analyzed for luciferase activities. Luciferase activity was normalized to the protein concentration. Data shown are means +/− SD of at least three experiments performed in quadruplicate (*** *p* < 0.001). Reproduced with modifications from ref. [48] with permission from Elsevier.

**Figure 4 cells-10-00875-f004:**
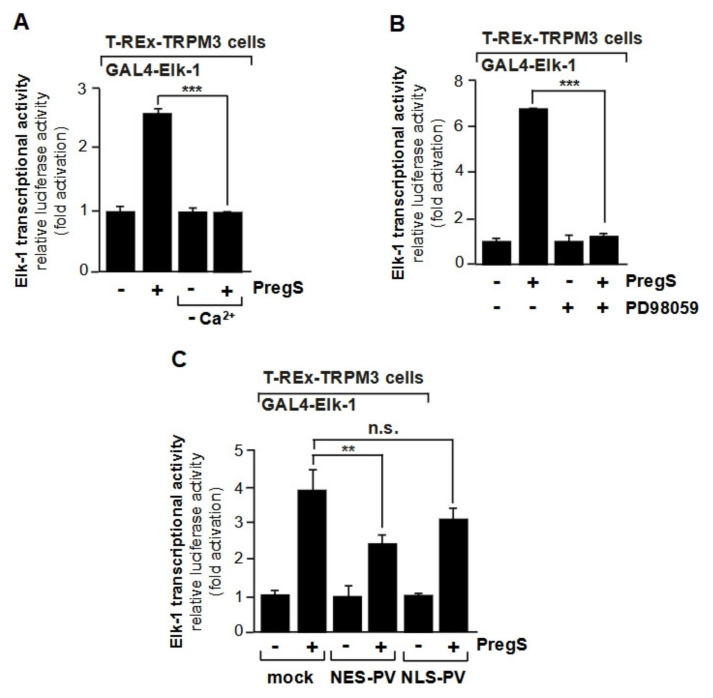
Stimulation of the TRPM3 Ca^2+^ channel increases the transcriptional activation potential of Elk-1, involving a rise of cytosolic Ca^2+^. (**A**) An influx of Ca^2+^ ions into the cells is essential for the activation of Elk-1 following stimulation of TRPM3 Ca^2+^ channels. HEK293 expressing TRPM3 Ca^2+^ channels were infected with a lentivirus containing a GAL4-responsive luciferase reporter gene (*UAS5Sp12.luc*). In addition, cells were infected with a lentivirus encoding GAL4-Elk-1 fusion protein, used to monitor the transcriptional activation potential of Elk-1. The cells were serum-starved for 24 h in either DMEM or Ca^2+^-free medium. Cells were stimulated with the TRPM3 ligand pregnenolone sulfate (PregS, 20 µM) for 24 h. Cell extracts were prepared and analyzed for luciferase activities. Luciferase activity was normalized to the protein concentration. (**B**) Extracellular signal-regulated protein kinase (ERK1/2) functions as a signal transducer connecting TRPM3 channels with the Elk-1 transcription factor. HEK293 cells expressing TRPM3 Ca^2+^ channels and the GAL4-Elk-1 fusion were infected with a recombinant lentivirus containing a GAL4-responsive reporter gene (*UAS^5^Sp1^2^.luc*). The cells were preincubated for 3 h with PD98059 (50 µM) and then stimulated with pregnenolone sulfate (PregS, 20 µM) for 24 h in the presence of PD98059. Cell extracts were prepared and analyzed for luciferase activities. (**C**) A rise in cytoplasmic Ca^2+^ is required for the upregulation of the transcriptional activation potential of Elk-1 following activation of TRPM3 Ca^2+^ channels. HEK293 cells expressing TRPM3 and GAL4-Elk-1 were infected with a lentivirus to insert the GAL4-responsive reporter gene UAS^5^Sp1^2^.luc into the chromatin. In addition, cells were infected with a lentivirus encoding either mCherry (control), NES-PVmCherry, or NLS-PVmCherry. The cells were serum-starved for 24 h and then stimulated with the TRPM3 ligand pregnenolone sulfate (PregS, 20 µM) for 24 h. Cell extracts were prepared and analyzed for luciferase activities. Data shown are mean +/− S.D. of three independent experiments performed in quadruplicate (** *p* < 0.001, n.s., not significant). Reproduced with modifications from ref. [14] with permission from Elsevier.

**Figure 5 cells-10-00875-f005:**
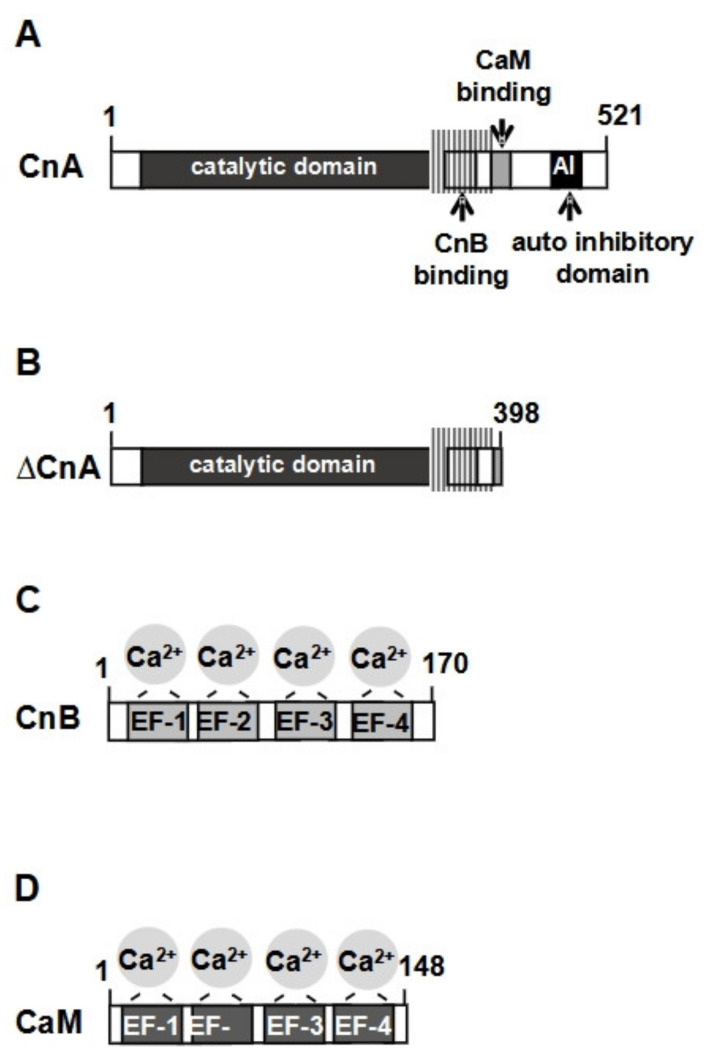
Modular structure of calcineurin. (**A**) Domain structure of calcineurin A. The enzyme has an extended catalytic domain, binding sites for calcineurin B and calmodulin and a *C*-terminal autoinhibitory domain that keeps the holoenzyme in an inactive state in the absence of elevated Ca^2+^ concentrations. (**B**) The truncated calcineurin A mutant (ΔCnA) lacks the autoinhibitory domain and a portion of the calmodulin binding site. (**C**,**D**) Calcineurin B (**C**) and calmodulin (**D**) are small proteins containing 4 EF hands for Ca^2+^ binding.

**Figure 6 cells-10-00875-f006:**
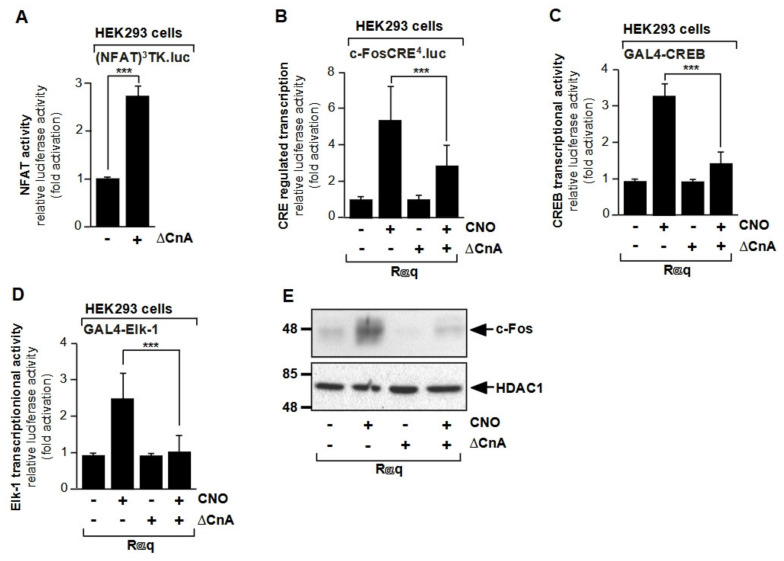
Regulation of gene transcription by calcineurin. (**A**) Expression of ΔCnA, a constitutively active mutant of calcineurin A, increased transcription of an NFAT-responsive reporter gene. HEK293 cells containing a chromatin-integrated NFAT-responsive reporter gene (NFAT)^3^TK.luc were infected with a recombinant lentivirus encoding ΔCnA. The cells were serum-starved for 24 h. Cell extracts were prepared and analyzed for luciferase activities. Luciferase activity was normalized to the protein concentration. (**B**) Calcineurin attenuates CRE-regulated gene transcription in HEK293 cells expressing activated Gαq-coupled designer receptors. HEK293 cells containing a chromatin-embedded CRE-containing reporter gene (c-FosCRE^4^.luc) were infected with a lentivirus encoding a Gαq-coupled designer receptor (Rαq). Cells were either mock-infected or additionally infected with a lentivirus encoding ΔCnA as indicated. The cells were serum-starved for 24 h and then stimulated with the Rαq ligand clozapine-*N*-oxide (CNO, 1 µM) for 24 h. Cell extracts were prepared and analyzed. (**C**) Expression of ΔCnA reduces the transcriptional activation potential of CREB in HEK293 cells expressing activated Gαq-coupled designer receptors. HEK293 cells that expressed the Gαq-coupled designer receptor Rαq and a GAL4-CREB fusion protein were used to measure the transcriptional activation potential of CREB. In addition, the cells contained a chromatin-integrated GAL4-responsive reporter gene used to measure the transcriptional activity of the GAL4-CREB fusion protein. Cells were either mock-infected or infected with a lentivirus encoding ΔCnA as indicated. Cells were stimulated with CNO (1 µM) for 24 h. (**D**) Expression of ΔCnA reduces the transcriptional activation potential of Elk-1 in HEK293 cells expressing activated Gαq-coupled designer receptors. HEK293 cells that expressed the Gαq-coupled designer receptor Rαq and a GAL4-Elk-1 fusion protein were used to measure the transcriptional activation potential of Elk-1. In addition, cells contained a chromatin-integrated GAL4-responsive reporter gene to measure the transcriptional activity of the GAL4-Elk-1 fusion protein. Cells were either mock-infected or infected with a lentivirus encoding ΔCnA as indicated. Cells were stimulated with CNO (1 µM) for 24 h. Luciferase activity was normalized to the protein concentration. Data shown are means +/− SD of at least three experiments performed in quadruplicate (*** *p* < 0.001). (**E**) Expression of a constitutively active mutant of calcineurin A (ΔCnA) negatively regulates c-Fos expression following stimulation of a Gαq-coupled designer receptor (Rαq). HEK293 cells expressing Rαq were either mock-infected or infected with a lentivirus encoding ΔCnA. Cells were cultured for 24 h in medium containing 0.05% serum and then stimulated with the Rαq ligand CNO (1 µM) for 3 h. Western blot analysis of proteins of nuclear extracts using an antibody directed against c-Fos. The antibody directed against HDAC1 was used as a loading control. Reproduced with modifications from ref. [68] with permission from Elsevier.

**Figure 7 cells-10-00875-f007:**
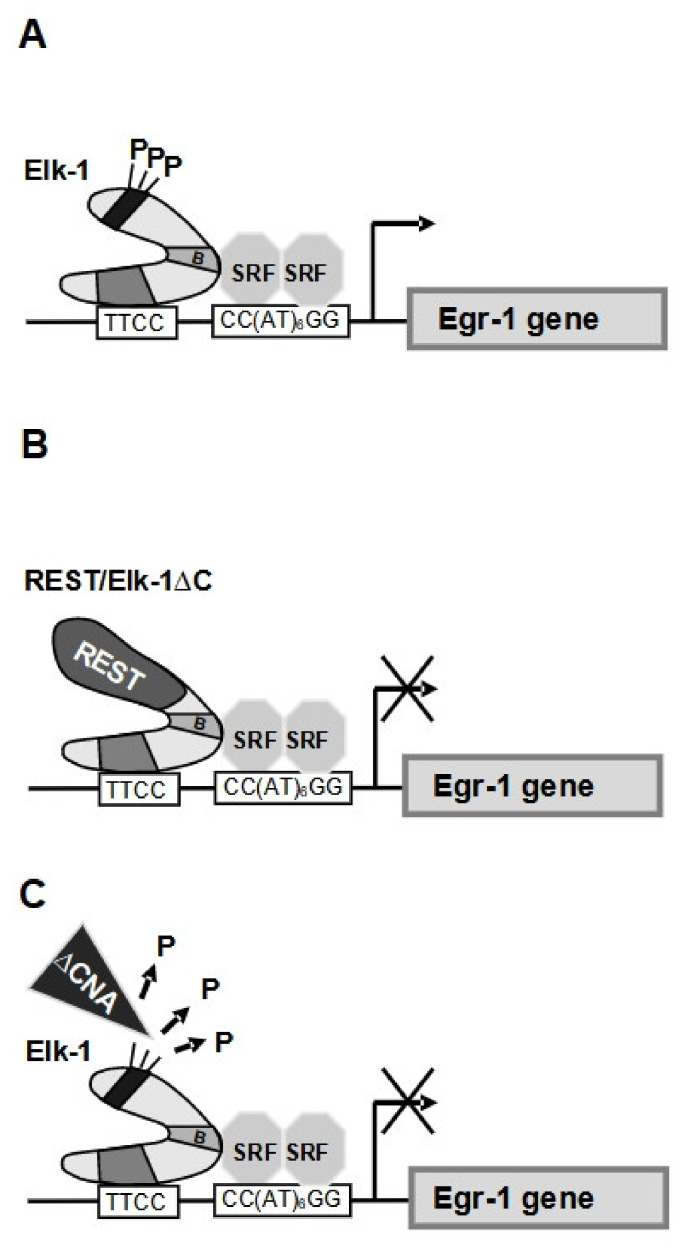
Orchestrated activities of calcineurin, Elk-1 and Egr-1 in pancreatic β-cells regulating islet size, apoptosis and glucose homeostasis. (**A**) The gene encoding the transcription factor Egr-**1** contains five serum-response elements (SREs) that serve as binding sites for Elk-1 and serum response factor (SRF). Transgenic mice expressing a constitutively active mutant of Egr-1, that interferes with Egr-1-regulated gene transcription in pancreatic β-cells, showed a phenotype with reduced islet size, impaired glucose tolerance and increased caspase-3 activity. (**B**) A similar phenotype was obtained in transgenic mice expressing a dominant-negative mutant of Elk-1. This mutant, named REST/Elk-1ΔC, inhibits expression of SRE-regulated genes, including the Egr-1-encoding gene. (**C**) Expression of a constitutively active mutant of calcineurin in pancreatic β-cells generated a similar phenotype. Calcineurin dephosphorylates and inactivates Elk-1. Reproduced with modifications from ref. [84] with permission from Elsevier.

**Table 1 cells-10-00875-t001:** Phenotypes of transgenic mice models.

Mouse Model	β-cell ↓ Mass	Hyperglycemia	↑ Apoptosis	Reference
Expression of a constitutively active mutant of **calcineurin** in β-cells	√	√	√	[83]
Expression of a dominant negative mutant of Elk-1 in β-cells	√	√	√	[84]
Expression of a dominant negative mutant of Egr-1 in β-cells	√	√	√	[85]

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
