# Peer review of "Ca2+ Microdomains, Calcineurin and the Regulation of Gene Transcription"

_cells, 2021, doi:10.3390/cells10040875_

Round 1

Reviewer 1 Report

This review provides a concise and much-needed summary of the field of calcium-stimulated transcription. The role of calcium-induced transcription factors has become particularly important in understanding the brain, but few overviews of the field exist. I think this is an excellent summary. My only major recommendation is to discuss more the interplay and relationship between different transcription factors, and to also include a table that summarizes all the isoforms of the various transcription factors that are calcium-regulated. A minor problem is that some corrections in the English would be helpful (e.g., 'uniformly' instead of 'uniformally' in the Abstract).

Author Response

April 5, 2021

[Cells] Manuscript ID: cells-1149811 

Dear Sir or Madam,

I would like to thank you for the examination of our manuscript. I hope that you find the revised version of the manuscript satisfactory and acceptable for publication in "Cells“.

Sincerely yours,

Gerald Thiel PhD

___________________________________________________________________

This review provides a concise and much-needed summary of the field of calcium-stimulated transcription. The role of calcium-induced transcription factors has become particularly important in understanding the brain, but few overviews of the field exist. I think this is an excellent summary. My only major recommendation is to discuss more the interplay and relationship between different transcription factors, and to also include a table that summarizes all the isoforms of the various transcription factors that are calcium-regulated. A minor problem is that some corrections in the English would be helpful (e.g., 'uniformly' instead of 'uniformally' in the Abstract).

The manuscript was checked by a native English speaker and many changes have been made throughout the text.

We discussed in the manuscript selected Ca2+-response transcription factors such as CREB, Elk-1 and NFAT. We had not the intention to list all known Ca2+-responsive transcription factors. In this case, a table would be helpful. There are some splice variants of CREB and Elk-1. We did not mention these isoforms in the text, because they are expressed only in particular tissues (e.g. expression of a CREB splice variant in testis) or the expression level is low and the functional implication are still not clear (CREB splice variant). I am not sure what the reviewer intended with the notion about the “interplay and relationship between different transcription factors”. There are genes that bind several Ca2+-response transcription factors. The c-Fos gene, for example, provides binding sites for CREB and Elk-1. However, these proteins bind to distinct DNA sites and they do not interact with each other.

Reviewer 2 Report

Ca2+ microdomains, calcineurin, and the regulation of gene transcription – Gerald Thiel, Tobias Schmidt, and Oliver G. Rössler

Cells

This review by Thiel et al. focuses on how changes in cytosolic and nuclear Ca2+ concentration affect the cellular genetic expression pattern, activating different Ca2+-responsive transcription factors and Ca2+/calmodulin-dependent protein phosphatases and kinases. Particular attention is paid to the role of calcineurin and how it modulates the most important transcription factors, such as NFAT, CREB, and Elk-1. Overall, the paper is organic both in form and in content making the topic easily accessible also to a general readership. Besides, by finely discussing the most recent findings in the field, the authors manage also to corroborate their belief that the amplitude and duration of the Ca2+ signal is the major element to affect gene transcription.

Some minor comments:

- For the sake of clarity, at the beginning of section 2 page 2, the authors could briefly define the MAPK family of protein kinases and its subfamilies, the ERKs, JNKs, and p38 kinases.

- Regarding page 3 “Low levels of Orai channel stimulation leads to an activation of NFAT….”: the actual definition of NFAT is given afterwards, at page 4. Thus, at page 3 the authors should not use the NFAT acronymous.

- Page 4 figure 2B: the authors should define in the figure caption the definition of the acronymous Ets and CArG.

- Page 6 figure 3, page 8 figure 4, and page 14 figure 6: the authors should report on each graph the statistics.

- Regarding page 9 “…. a low stimulus intensity also triggers NFAT and NFAT4”, please correct to “…. a low stimulus intensity also triggers NFAT1 and NFAT4”,

- For the sake of clarity, the authors should rephrase at page 10 the following sentences, avoiding redundancy: “Calcineurin became of medical interest following the discovery that the clinically im-portant immunosuppressive drugs tacrolimus (FK506) and cyclosporin A, used for the treatment of organ rejection in transplanted patients, function as potent calcineurin inhibitors when they are bound to the immunophilins FKBP12 or cycophilin A, respectively. Both compounds make complexes with the immunophilins cyclophilin A and FKBP12”.

- Regarding page 11 “There are several genetically encoded calcineurin inhibitors that may more suitable for investigation of calcineurin functions.”, please correct to “There are several genetically encoded calcineurin inhibitors that may be more suitable for investigation of calcineurin functions.”

Author Response

April 5, 2021

[Cells] Manuscript ID: cells-1149811 

Dear Sir or Madam,

I would like to thank you for the examination of our manuscript. I hope that you find the revised version of the manuscript satisfactory and acceptable for publication in "Cells“.

Sincerely yours,

Gerald Thiel PhD

___________________________________________________________________

This review by Thiel et al. focuses on how changes in cytosolic and nuclear Ca2+ concentration affect the cellular genetic expression pattern, activating different Ca2+-responsive transcription factors and Ca2+/calmodulin-dependent protein phosphatases and kinases. Particular attention is paid to the role of calcineurin and how it modulates the most important transcription factors, such as NFAT, CREB, and Elk-1. Overall, the paper is organic both in form and in content making the topic easily accessible also to a generalreadership. Besides, by finely discussing the most recent findings in the field, the authors manage also to corroborate their belief that the amplitude and duration of the Ca2+ signal is the major element to affect gene transcription.

Some minor comments:

- For the sake of clarity, at the beginning of section 2 page 2, the authors could briefly define the MAPK family of protein kinases and its subfamilies, the ERKs, JNKs, and p38 kinases.

We defined the MAP kinases in the revised version of the manuscript.

- Regarding page 3 “Low levels of Orai channel stimulation leads to an activation of NFAT….”: the actual definition of NFAT is given afterwards, at page 4. Thus, at page 3 the authors should not use the NFAT acronymous.

We explained the abbreviation NFAT on page 3

- Page 4 figure 2B: the authors should define in the figure caption the definition of the acronymous Ets and CArG.

We explained the abbreviations Ets and CArG in the legend to figure 2.

- Page 6 figure 3, page 8 figure 4, and page 14 figure 6: the authors should report on each graph the statistics.

Statistics is included in the graphs.

- Regarding page 9 “…. a low stimulus intensity also triggers NFAT and NFAT4”, please correct to “…. a low stimulus intensity also triggers NFAT1 and NFAT4”,

The sentence has been corrected as requested.

- For the sake of clarity, the authors should rephrase at page 10 the following sentences, avoiding redundancy: “Calcineurin became of medical interest following the discovery that the clinically important immunosuppressive drugs tacrolimus (FK506) and cyclosporin A, used for the treatment of organ rejection in transplanted patients, function as potent calcineurin inhibitors when they are bound to the immunophilins FKBP12 or cycophilin A, respectively. Both compounds make complexes with the immunophilins cyclophilin A and FKBP12”.

We deleted the last sentence in the revised manuscript.

- Regarding page 11 “There are several genetically encoded calcineurin inhibitors that may more suitable for investigation of calcineurin functions.”, please correct to “There are several genetically encoded calcineurin inhibitors that may be more suitable for investigation of calcineurin functions.”

We corrected the sentence.

In addition, the manuscript was checked by a native English speaker and many changes have been made throughout the text.

Reviewer 3 Report

Revision: Ca2+ microdomains, calcineurin, and the regulation of gene transcription
The study by Thiel et al. focuses on Ca2+ dependent gene regulation in the context of active calcineurin. The authors describe the change of Ca2+ levels in the cytosol and in the nucleus and link it to specific activation of transcription factors, like NFAT, ELK1 or CREB. Activation of these transcription factors under several conditions has been studied in luciferase assays. In general, this is a highly interesting topic and it is of high importance for the field.

In this work several main aspects of Ca2+ signaling pathways are investigated thoroughly, however, I would suggest to include the following points to get an even more complete picture of the signaling pathways.

1. Considering the title of the manuscript, the authors may include a more detailed description of Ca2+ microdomains, in size, concentration and localization.
2. Especially depletion of lysosomal Ca2+ stores will be interesting, as they have been shown to lead to Calcineurin activation and subsequent transcription factor activation (Medina et al, Nat Cell Biol. 2015).
3. As in the article NFAT activation is described extensively, I would recommend to include also Orai channels in figure 1. Especially as IP3 dependent store‐depletion of the ER is also depicted.
4. I would suggest to also include a detailed description about Calmodulin. It is a crucial part of Ca2+ signaling and is mostly important for your study due to Calcineurin activation.
5. It should be mentioned that also other transcription factors are regulated in a Ca2+ dependent manner, like HLH TFs SEF1, TFEB and MYC.
6. For completeness, Ca2+ dependent NFAT activation could be described in more detail, focusing on specific calcium oscillations and Orai isoforms. Please consider the following publications: Kar et al. Mol Cell. 2016 & Yoast et al. Nat Commun. 2020.
7. Regarding Fig. 4. How does the Ca2+ level in the cell change in the absence of Ca2+ for 24h in comparison to normal conditions?
8. Please make sure the meaning of abbreviations is explained the first time when its mentioned (see NFAT).
9. “Overexpression of RCAN1 has been found in the brain of Alzheimer´s disease patients and humans with Down syn‐drome. The RCAN proteins are thought to establish a negative feedback loop that pre‐vents high calcineurin activity.” Please add reference.
10. Please check your grammar and spelling, especially in the following sentences:
“These enzymes function as signal transducers for propagate the Ca2+ signal from the cytosol into the nucleus.”
“Calcineurin has also been shown to regulate the Raf/ERK1/2 signaling cascade by dephosphoryating B‐Raf protein kinase.”
“A significant translocation of calcineurin into the nucleus has been oberved after stimulation of RBL mast cells with thapsigargin”.
“There are several genetically encoded calcineurin inhibitors that may more suitable for investigation of calcineurin functions.”
“NFAT transcription factors are cytoplasmic phosphoproteins that translocate into the nucleus upon dephosphorylation by the calcineurin.”
11. Please add the isoform number to the second last NFAT, I think it should be NFAT1. “Viewing the entire signal cascade, these results suggest that a high stimulus intensity triggers the nuclear translocation of both NFAT1 and NFAT4 and the sustained expression of both proteins in the nucleus, whereas a low stimulus intensity also triggers NFAT and NFAT4 translocation, but NFAT4 is rapidly exported back into the cytoplasm.”

Author Response

April 5, 2021

[Cells] Manuscript ID: cells-1149811

Dear Sir or Madam,

I would like to thank you for the examination of our manuscript. We have dealt with the criticisms and comments. Please find below a point-by-point response to the points of criticism raised. I hope that you find the revised version of the manuscript satisfactory and acceptable for publication in "Cells“.

Sincerely yours,
Gerald Thiel, PhD ___________________________________________________________________

Revision: Ca2+ microdomains, calcineurin, and the regulation of gene transcription.

The study by Thiel et al. focuses on Ca2+ dependent gene regulation in the context of active calcineurin. The authors describe the change of Ca2+ levels in the cytosol and in the nucleus and link it to specific activation of transcription factors, like NFAT, ELK1 or CREB. Activation of these transcription factors under several conditions has been studied in luciferase assays. In general, this is a highly interesting topic and it is of high importance for the field.

In this work several main aspects of Ca2+ signaling pathways are investigated thoroughly, however, I would suggest to include the following points to get an even more complete picture of the signaling pathways.

1. Considering the title of the manuscript, the authors may include a more detailed description of Ca2+ microdomains, in size, concentration and localization.

We discussed microdomains generated by the influx of Ca2+ through plasma membrane channels and intracellular Ca2+ channels. We mentioned the change of the Ca2+ concentration in the cytoplasm by Ca2+ accumulating and Ca2+ releasing organelles, Ca2+ pumps and Ca2+ exchangers. A discussion of the size, concentration and localization of Ca2+ microdomains in detail would be the topic of a separate review that should also critically view the methods to determine intracellular free Ca2+ concentrations.

2. Especially depletion of lysosomal Ca2+ stores will be interesting, as they have been shown to lead to Calcineurin activation and subsequent transcription factor activation (Medina et al, Nat Cell Biol. 2015).

The release of Ca2+ ions from lysosomes was already mentioned in the “Introduction” section:

“Even lysosomes, often viewed as a “terminal end” of intracellular pathways, are involved in intracellular Ca2+ signaling.” We also mentioned in the manuscript the activation of calcineurin following a Ca2+ release from lysosomes.

3. As in the article NFAT activation is described extensively, I would recommend to include also Orai channels in figure 1. Especially as IP3 dependent store-depletion of the ER is also depicted.

We included the Orai channel in Fig. 1 and discussed Orai signaling in the text in the context of a Ca2+ microdomain near the open Orai channel.

4. I would suggest to also include a detailed description about Calmodulin. It is a crucial part of Ca2+ signaling and is mostly important for your study due to Calcineurin activation.

I agree with the reviewer that calmodulin is a very important part of Ca2+ signaling. However, calmodulin has many functions in the cell that a detailed discussion would not fit into the review addressing Ca2+-regulated gene transcription. We added in Fig. 5 a cartoon of calmodulin, showing the similar modular structure of calmodulin and calcineurin B.

5. It should be mentioned that also other transcription factors are regulated in a Ca2+ dependent manner, like HLH TFs SEF1, TFEB and MYC.

We focused the review article on the discussion of the Ca2+-responsive transcription factors CREB, NFAT, and Elk-1, because many data about these proteins are available and hypotheses have been stated about the cooperation between Ca2+ ions in different microdomains, calcineurin, and the activation of these transcription factors. Certainly, TFEB is an interesting protein, but we decided not to include this protein in the article.

6. For completeness, Ca2+ dependent NFAT activation could be described in more detail, focusing on specific calcium oscillations and Orai isoforms. Please consider the following publications: Kar et al. Mol Cell. 2016 & Yoast et al. Nat Commun. 2020.

We mentioned the connection between nuclear translocation of NFAT1 and the Ca2+ microdomain, generated by the Ca2+ influx via Orai channels

7. Regarding Fig. 4. How does the Ca2+ level in the cell change in the absence of Ca2+ for 24h in comparison to normal conditions?

We did not measure the change of the Ca2+ concentration in insulinoma cells cultured for 24 h in Ca2+-free medium. Rather, we showed that this treatment blocked glucose-induced transcription (Müller et al., Cell Calcium 52 (2012).

8. Please make sure the meaning of abbreviations is explained the first time when its mentioned (see NFAT).

We explained the abbreviation NFAT the first time when it was mentioned.

9. “Overexpression of RCAN1 has been found in the brain of Alzheimer ́s disease patients and humans with Down syn-drome. The RCAN proteins are thought to establish a negative feedback loop that pre-vents high calcineurin activity.” Please add reference.

References were added as requested.

10. Please check your grammar and spelling, especially in the following sentences:

“These enzymes function as signal transducers for propagate the Ca2+ signal from the cytosol into the nucleus.”

“Calcineurin has also been shown to regulate the Raf/ERK1/2 signaling cascade by dephosphoryating B-Raf protein kinase.”

“A significant translocation of calcineurin into the nucleus has been oberved after stimulation of RBL mast cells with thapsigargin”.

“There are several genetically encoded calcineurin inhibitors that may more suitable for investigation of calcineurin functions.”

“NFAT transcription factors are cytoplasmic phosphoproteins that translocate into the nucleus upon dephosphorylation by the calcineurin.”

All sentences have been rephrased or deleted. In addition, the manuscript was checked by a native English speaker and many changes have been made throughout the text.

11. Please add the isoform number to the second last NFAT, I think it should be NFAT1. “Viewing the entire signal cascade, these results suggest that a high stimulus intensity triggers the nuclear translocation of both NFAT1 and NFAT4 and the sustained expression of both proteins in the nucleus, whereas a low stimulus intensity also triggers NFAT and NFAT4 translocation, but NFAT4 is rapidly exported back into the cytoplasm.”

We corrected the sentence.